# Factors influencing the outcomes of non-pharmacological interventions for managing fatigue across the lifespan of people living with musculoskeletal (MSK) conditions: a scoping review protocol

Katie Fishpool [1,2] George Young,[1] Coziana Ciurtin [3] Fiona Cramp [1] Emmanuel Oghenetejiri Erhieyovwe,[4] Bayram Farisogullari,[3] Gary J Macfarlane,[5] Pedro M Machado [6,7] Jen Pearson [1] Eduardo Santos,[8] Emma Dures [1,2]

For numbered affiliations see end of article.

**Correspondence to**
Katie Fishpool;
katie.fishpool@uwe.ac.uk

## ABSTRACT

**Introduction** Fatigue is an important and distressing symptom for many people living with chronic musculoskeletal (MSK) conditions. Many non-pharmacological interventions have been investigated in recent years and some have been demonstrated to be effective in reducing fatigue and fatigue impact, however, there is limited guidance for clinicians to follow regarding the most appropriate management options. The objective of this scoping review is to understand and map the extent of evidence in relation to the factors that relate to the outcome of non-pharmacological interventions on MSK condition-related fatigue across the lifespan.

**Methods and analysis** This scoping review will include evidence relating to people of all ages living with chronic MSK conditions who have been offered a non-pharmacological intervention with either the intention or effect of reducing fatigue and its impact. Databases including AMED, PsycINFO, CINAHLPlus, MEDLINE, EMBASE and Scopus will be searched for peer-reviewed primary research studies published after 1 January 2007 in English language. These findings will be used to identify factors associated with successful interventions and to map gaps in knowledge.

**Ethics and dissemination** Ethical approval was not required for this review. Findings will be disseminated by journal publications, conference presentations and by communicating with relevant healthcare and charity organisations.

## STRENGTHS AND LIMITATIONS OF THIS STUDY

⇒ Patient and public involvement and engagement workshops at key time points will ensure that the protocol, review findings and subsequent discussion are relevant to stakeholders and reflect lived experience of musculoskeletal fatigue.
⇒ All studies will be reviewed, and data extraction will be checked by a minimum of two researchers.
⇒ The effectiveness of specific interventions and methodological quality of included studies is not covered in this scoping review.
⇒ Only evidence available in English will be reviewed.

## INTRODUCTION

Musculoskeletal (MSK) conditions include inflammatory and non-inflammatory conditions such as connective tissue diseases, inflammatory arthritis and osteoarthritis, back and neck pain, and fibromyalgia which affect the muscles, bones, joints and connective tissue.[1 2] More than 10 million people in the UK and 1.7 billion people globally currently live with an MSK condition.[1 2] Prevalence increases with age but these conditions are encountered across the lifespan, with approximately 234 000 children in England and Scotland living with a long-term MSK condition.[2]

Fatigue has been identified by people living with chronic MSK conditions as a priority symptom which has a significant impact on quality of life.[2–8] Pharmacological treatments are not licensed for the management of fatigue without concurrent disease activity, so the focus in clinical practice has been on non-pharmacological options.[9 10] This has been mirrored in healthcare research and recent systematic reviews have examined the strength of the evidence supporting a range of non-pharmacological interventions in different patient groups.[11–15] Non-pharmacological interventions are any non-chemical or biological interventions that are theoretically based and empirically

proven, or that have a logical rationale which is possible to prove by empirical study.[11 16]

Recent and ongoing studies on the topic of fatigue support the need for this scoping review and have been used to inform its design. A previous scoping review exploring fatigue in patients with rheumatic and MSK conditions[17] reported on the efficacy of interventions and also considered determinants associated with fatigue. A systematic review assessed the quality of evidence available to support non-pharmacological interventions to reduce fatigue in patients with inflammatory rheumatic conditions.[14] The National Institute for Health and Care Research has recently awarded funding for the project 'Effectiveness of Interventions For FatiguE in Long term conditions' (EIFFEL).[18] Study protocols have been registered on the PROSPERO database which state that the team will be conducting two systematic reviews; one to assess the effectiveness of interventions for fatigue in long-term conditions[19] and the other to explore the acceptability of interventions for fatigue.[20] There is potential for duplication of effort when considering acceptability of interventions, as the qualitative data may explore contextual factors and some musculoskeletal conditions are also included in their searches. However, the proposed methodology of this study is a scoping review which allows for a broader view of contextual factors across the lifespan of all chronic MSK conditions and will be gathered from multiple study designs, which is appropriate for the aims of the review to map existing knowledge. Contact has been made with the EIFFEL team to reduce the risk of duplication and share knowledge.

The European Alliance of Associations for Rheumatology (EULAR) recently funded a taskforce to examine the effectiveness of non-pharmacological interventions for fatigue in inflammatory rheumatic conditions and the resulting systematic review found strong evidence that some interventions are effective.[14] This informed the EULAR recommendations for the management of fatigue[21] and highlighted a need to better understand contextual factors and the mechanisms by which interventions are effective.[7] There is currently no comprehensive understanding of the factors which influence the success of an intervention. The impact of this is a lack of evidence to support decision making in how to design, offer and deliver interventions in the most effective way, tailored to a range of patients and at the optimal time. Current clinical guidelines for the management of common MSK conditions recognise fatigue as an important symptom but do not make any recommendations for how it can be addressed directly,[22–24] hindering the implementation of the evidence.

The clinical pathway for the management of MSK conditions in the UK differs depending on primary diagnosis, with suspected inflammatory conditions being referred to specialist secondary care settings and osteoarthritis and fibromyalgia being managed predominantly through primary care.[24–26] The impact of this is that the experience of patients with MSK fatigue and the profession and skills of the clinicians who provide their care may be quite different.

## Review aims

The aim of this scoping review is to generate evidence for health professionals and educators to design or adapt tailored MSK-fatigue support. The objectives of this review are to identify evidence for existing interventions for MSK fatigue across the life course and to explore the theoretical basis for the interventions. To explore the comprehensive nature of the existing evidence, the clinical and demographic characteristics of the participants as well as to capture the training/skills of those who deliver the interventions where this information is available. As the intention is to create an overview of the current knowledge and to highlight gaps in the existing literature rather than to assess the effectiveness of specific interventions, a scoping review is the most appropriate methodological approach.[27 28]

## METHODS AND ANALYSIS

### Study design

In accordance with the Joanna Briggs Institute (JBI) methodology for scoping reviews,[29] this protocol sets out the criteria that the reviewing team will use to include and exclude sources of evidence and to identify what data is relevant. The data will be reported using the Preferred Reporting Items for Systematic Reviews and Meta Analyses (PRISMA) extension for scoping reviews.[30]

### Identifying relevant studies

A preliminary search of MEDLINE in August 2023 identified the current scope and scale of the evidence base related to the scoping review. The search strategy (online supplemental appendix 1) was then developed with support from a specialist subject librarian and reviewed by stakeholders in a patient and public involvement and engagement (PPIE) workshop.

The following electronic databases will be searched for research published in peer-reviewed journals from January 2007 onwards; AMED, MEDLINE, PsycINFO, CINAHLPlus (EBSCO platform), EMBASE (Ovid platform), SCOPUS and Cochrane Database. This date was chosen to correspond with the Outcome Measures in Rheumatology 8th meeting (OMERACT 8) which endorsed fatigue as an addition to the 'core set' of outcome measures for all subsequent studies, highlighting the importance of investigating this symptom.[31] There will be no restrictions on the age of participants, allowing interventions that have been used throughout the life course and highlighting any gaps in provision. A search for unpublished studies will not be conducted due to the limitations of time to complete this review. The reference lists from included studies will be hand searched to check for any other relevant papers not captured in the database searching. Only evidence available in English will be reviewed and studies in other languages will be excluded

**Table 1** Inclusion and exclusion criteria

| Inclusion criteria | Exclusion criteria |
|---|---|
| ▸ Primary research study<br>▸ Published in a peer-reviewed journal<br>▸ Available in English language<br>▸ Participants have one or more chronic MSK conditions<br>▸ Participants experience fatigue at baseline<br>▸ Published during or after 2007<br>▸ Describes a non-pharmacological intervention to manage MSK condition symptoms, with fatigue reduction as a primary or secondary outcome | ▸ Reviews, protocols, opinion pieces, editorials, case series, case reports, observational cohort studies<br>▸ Pharmacological interventions<br>▸ No intervention is described<br>▸ Muscle fatigue rather than global fatigue is examined<br>▸ No data is available on factors associated with intervention success (theoretical mechanism of intervention OR characteristics of participants OR characteristics of clinicians delivering interventions) |

MSK, musculoskeletal.

due to time. All studies will be uploaded to the review software system Covidence.[32]

### Selection of studies

A minimum of two independent reviewers will screen the titles and abstracts of all identified studies against the stated inclusion and exclusion criteria (table 1). Regular meetings will be held by the team throughout the title and abstract screening process to aid understanding and reduce disagreements.[33] Papers that proceed to the full-text stage of screening will also be reviewed by two or more independent reviewers who will document the main reason any excluded papers do not meet the inclusion criteria. Any differences in opinion between the reviewers will be resolved by discussing the papers, with an additional independent reviewer to support mediation, as required.

All primary research methodologies will be considered, including experimental and quasi-experimental study designs, before and after studies and interrupted time series studies, analytical observational studies, case-control studies and analytical cross-sectional studies, and descriptive observational study designs. Qualitative studies will also be considered including, for example, phenomenology, grounded theory, ethnography, qualitative description and action research.

### Data extraction

Data from included papers will be extracted by one reviewer and checked by another using an adapted version of the JBI template for data extraction (online supplemental appendix 2). This template captures information about study participants, methods and findings relevant to the research question[34] and has been amended

to extract additional data on contextual factors including clinical characteristics, information on clinicians delivering interventions and the hypothesis behind the design of the intervention. Any disagreements that may arise between the reviewers will be resolved by consensus, with an additional reviewer to support mediation. If necessary, the authors of the included papers will be contacted for further information or data clarification.

### Data reporting and analysis

Findings will be presented in a PRISMA flow diagram[35] to demonstrate the number of articles identified and their sources, with reasons for exclusion at full-text screening summarised. All included studies will be summarised in tabular format. Further, figures will be used to illustrate a map of the existing literature and any gaps highlighted by the review. Data analysis is likely to be narrative due to the broad range of study types being included. This may be further refined for use during the review process.

### Patient and public involvement statement

A PPIE workshop was held in October 2023 during the design stage of the search strategy and review protocol. Stakeholders attending the workshop included patients living with one or more MSK conditions and clinicians from a range of professions who support patients experiencing MSK-related fatigue. The discussion focused on people's experiences of offering or being offered support to manage their fatigue and asking for comments on the proposed review. This highlighted additional intervention types and pathways that were subsequently included in the search terms (online supplemental appendix 3). It also confirmed our understanding that fatigue is a significant issue and that an overview of potential interventions and management techniques would be welcomed by patients and clinicians.

Further, PPIE events are planned at key points during the project to ensure the validity of the final review. Workshops that focus on the support of adults and of children and young people will be held to discuss the initial findings following data extraction with the aim of highlighting key themes and gaps in knowledge. A further pair of workshops will be arranged following the synthesis of the findings to ensure the validity of the review, discuss priorities for future research and promote dissemination of the findings through appropriate groups. The outcome of these events and how they influence the scoping review process will be reported in the final review following the guidelines for the GRIPP2 short form reporting checklist,[36] which is a tool designed to improve the reporting of public and patient involvement in research.

### ETHICS AND DISSEMINATION

Ethical approval is not required for this scoping review. The findings of this review will be disseminated via relevant peer-reviewed journals, conference presentations

and through sharing findings with relevant charities and health professionals.

## Author affiliations
[1] School of Health and Social Wellbeing, University of the West of England, Bristol, UK
[2] Academic Rheumatology, University Hospitals Bristol and Weston NHS Foundation Trust, Bristol, UK
[3] University College London, London, UK
[4] Hywel Dda University Health Board, Carmarthen, UK
[5] Aberdeen Centre for Arthritis and Musculoskeletal Health, University of Aberdeen, Aberdeen, UK
[6] Centre for Rheumatology, University College London, London, UK
[7] Neuromuscular Diseases, University College London, London, UK
[8] Hospitals of Coimbra University, Coimbra, Portugal

**Acknowledgements** The development of this scoping review search strategy was supported by Specialist Subject Librarian Pauline Shaw from the University of the West of England library. The review team is also grateful for the contributions of the stakeholders who attended the Patient and Public Involvement and Engagement workshop.

**Contributors** Project funding application by ED with support from FC, JP, GJM, PMM, CC, BF and ES. Drafting of protocol and search strategy by KF in discussion with ED and GY, reviewed by EOE, FC, JP, GJM, PMM, CC, BF and ES.

**Funding** This scoping review protocol is part of the project 'MusculoskelEtal faTigue acRoss the lIfe CourSe: understanding what helps and mapping what is missing (METRICS)' (reference 23140), which is jointly funded by Versus Arthritis and The Kennedy Trust.

**Competing interests** PMM has received consulting/speaker's fees from Abbvie, BMS, Celgene, Eli Lilly, Janssen, MSD, Novartis, Orphazyme, Pfizer, Roche and UCB, all unrelated to this project. There are no competing interests in this project.

**Patient and public involvement** Patients and/or the public were involved in the design, or conduct, or reporting, or dissemination plans of this research. Refer to the Methods section for further details.

**Patient consent for publication** Not applicable.

**Provenance and peer review** Not commissioned; externally peer reviewed.

**ORCID iDs**
Katie Fishpool http://orcid.org/0009-0000-7260-9577
Coziana Ciurtin http://orcid.org/0000-0002-8911-4113
Fiona Cramp http://orcid.org/0000-0001-8035-9758
Pedro M Machado http://orcid.org/0000-0002-8411-7972
Jen Pearson http://orcid.org/0000-0002-5754-2762
Emma Dures http://orcid.org/0000-0002-6674-8607

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
