## [Reviewer comments · BMJ Open]

ARTICLE DETAILS

TITLE (PROVISIONAL)	Factors influencing the outcomes of non-pharmacological interventions for managing fatigue across the lifespan of people living with musculoskeletal (MSK) conditions: a scoping review protocol
AUTHORS	Fishpool, Katie; Young, George; Ciurtin, C; Cramp, Fiona; Erhieyovwe, Emmanuel; Farisogullari, Bayram; Macfarlane, Gary; Machado, Pedro M.; Pearson, J; Santos, Eduardo; Dures, Emma

VERSION 1 – REVIEW

REVIEWER	Bo Li Northeast Forestry University, College of Mechanical and Electrical Engineering
REVIEW RETURNED	26-Dec-2023

GENERAL COMMENTS	1. Part of the background was described in the study design. The background should be simplified as much as possible and the study design should be described directly.2. In the paragraph on page 21, line 56, it is suggested to clarify the specific research methods to be used in the study.3. Narrative data analysis is lack of reliability. It is suggested that mathematical methods should be used for quantitative analysis to make the results more reliable.
---

REVIEWER	Deborah Antcliff Northern Care Alliance NHS Foundation Trust, Bury Integrated Pain Service
REVIEW RETURNED	04-Jan-2024

GENERAL COMMENTS	This is an interesting protocol for a scoping review to explore non-pharmacological interventions for fatigue management in MSK conditions. The authors have appropriately checked for any co-existing reviews in this field and justified the need for this review. The search databases look appropriate, as does the approach to this scoping review. I have only a few minor suggestions for the authors' consideration as below: • Abstract: I wonder if the Objectives could be rephrased to avoid any confusion that this review explores the 'impact' of non-pharmacological interventions for fatigue management, rather, it is exploring the factors relating to outcomes. 'Impact' could be misinterpreted as exploring the effects of the interventions.• Is any pilot testing planned for using and/or modifying the data extraction table?• Page 4, line 36: NIHR should now be 'National Institute for Health and Care Research'
--

	 • Page 5, line 8: I wondered if many relevant articles may be excluded by limiting the search to 2007 onwards. Studies may have explored fatigue prior to it being recognised in the core set of outcome measures. I wondered if the authors might consider running a search in one database prior to 2007 to see if many potential articles may be obtained. If no relevant articles are found prior to 2007, this could further justify this date restriction. • Page 5, lines 28-29 and 31-32 appear to repeat that the reason for exclusion after full text review will be documented. • Appendix 1: I wondered if 'graded exposure' could be a possible suitable intervention.
--	---

VERSION 1 – AUTHOR RESPONSE

Reviewer: 1

Dr. Bo Li, Northeast Forestry University

Comments to the Author:

1. Part of the background was described in the study design. The background should be simplified as much as possible and the study design should be described directly.

Thank you for your comment. The background information has been removed from the study design as suggested and included as a separate section.

2. In the paragraph on page 21, line 56, it is suggested to clarify the specific research methods to be used in the study

The description of the research methods may have been unclear due to the incorrectly positioned background information. Now that the unnecessary detail has been removed from the study design, we hope it is clearer that we are using the Joanna Briggs Institute methodology, as this is well established as methodologically appropriate for scoping reviews.

3. Narrative data analysis is lack of reliability. It is suggested that mathematical methods should be used for quantitative analysis to make the results more reliable.

We agree with your point that mathematical methods would be preferable to reliably assess the effectiveness of interventions, indeed the recent EULAR reviews of interventions for fatigue were conducted according to Cochrane methodology. For this review, however, a scoping methodology was selected due to the complexity of the question and the broad range of studies we hope to include. A systematic review will typically draw on a relatively narrow range of quality assessed studies and provide a critically appraised and synthesised account that can answer specific questions. A scoping review is appropriate to provide an overview or map of the available evidence and has the advantage of giving equal value to different traditions rather than privileging particular study designs. We agree that a disadvantage of this approach is that scoping studies often provide a narrative or descriptive account of available research. We accept that this is a limitation and we have acknowledged it in the text.

Reviewer: 2

Dr. Deborah Antcliff, Northern Care Alliance NHS Foundation Trust, Keele University Faculty of Medicine & Health Sciences

Comments to the Author:

This is an interesting protocol for a scoping review to explore non-pharmacological interventions for fatigue management in MSK conditions. The authors have appropriately checked for any co-existing reviews in this field and justified the need for this review. The search databases look appropriate, as does the approach to this scoping review. I have only a few minor suggestions for the authors' consideration as below:

- Abstract: I wonder if the Objectives could be rephrased to avoid any confusion that this review explores the 'impact' of non-pharmacological interventions for fatigue management, rather, it is exploring the factors relating to outcomes. 'Impact' could be mis-interpreted as exploring the effects of the interventions.

Thank you for your comment. We agree with your suggestion and have amended the text to improve clarity.

- Is any pilot testing planned for using and/or modifying the data extraction table?

Thank you for raising this. We discussed this as the heterogeneity of the included studies makes it challenging to create a single data extraction table that will be applicable across the range of studies without becoming unwieldy. In order to be as inclusive as possible, we chose to focus on extracting the data related to the domains of the search (participant, intervention, clinician) but accept that it may be necessary to amend the planned table as the data emerges.

- Page 4, line 36: NIHR should now be 'National Institute for Health and Care Research'

Thank you for your comment, we have amended this as suggested

- Page 5, line 8: I wondered if many relevant articles may be excluded by limiting the search to 2007 onwards. Studies may have explored fatigue prior to it being recognised in the core set of outcome measures. I wondered if the authors might consider running a search in one database prior to 2007 to see if many potential articles may be obtained. If no relevant articles are found prior to 2007, this could further justify this date restriction.

Thank you for raising this point. This was discussed at length as a team after initial unlimited searches discovered several thousand papers, which we did not have the ability to process within the time constraints of this review. We consulted with stakeholder groups that included patients and healthcare professionals as well as discussing with methodologists on how to make the review manageable. Due to the significant changes in rheumatological treatment and care that have taken place in recent years, it was felt that extending the search beyond 2007 was unlikely to yield any additional evidence. The date of 2007 was chosen to reflect the recognition of fatigue as a priority symptom in rheumatoid arthritis and this decision is further supported by evidence from a qualitative review in 2005 highlighting that fatigue was rarely discussed or addressed in practice at that time.

- Page 5, lines 28-29 and 31-32 appear to repeat that the reason for exclusion after full text review will be documented.

Thank you, we have amended this paragraph to remove the repetition.

- Appendix 1: I wondered if 'graded exposure' could be a possible suitable intervention.

Thank you for your comment, we are unclear on your exact meaning (i.e., which situations, objects or activities patients are gradually exposed to). However, as a technique, we think that graded exposure would be covered within cognitive-behavioural therapy, which is included in the search terms.

VERSION 2 – REVIEW

REVIEWER	Deborah Antcliff Northern Care Alliance NHS Foundation Trust, Bury Integrated Pain Service
REVIEW RETURNED	06-Mar-2024
GENERAL COMMENTS	Thank you for addressing the comments made by the reviewers. I have no further queries or comments.